# Mean-Field Game Equilibria in Score-Based Diffusion Models: Convergence Rates, Nash Stability, and Adversarial Robustness

## Abstract

We formulate the training of score-based diffusion models as a mean-field game (MFG) where an infinite population of denoising agents compete to minimize individual loss while collectively approximating the data distribution. This game-theoretic lens yields three main contributions. First, we prove existence and uniqueness of Nash equilibria for the MFG system under mild regularity conditions on the data distribution, establishing convergence at rate $O(1/N)$ in 2-Wasserstein distance as the number of discretization steps $N$ grows. Second, we derive a Hamilton-Jacobi-Bellman (HJB) equation governing the optimal score function and show that classical DDPM/DDIM training implicitly solves a forward-backward SDE system that characterizes the MFG equilibrium. Third, we prove that MFG equilibria are structurally stable against adversarial perturbations, providing the first game-theoretic robustness guarantee for diffusion models. We introduce MFG-Diffusion, a training algorithm that explicitly computes equilibrium strategies, demonstrating 8–12% improvement in FID on CIFAR-10 and CelebA-HQ-256, with provable convergence guarantees. Our framework unifies score matching, denoising diffusion, and flow matching under a single game-theoretic umbrella and opens new avenues for defending against adversarial perturbations in diffusion-based systems.

## 1 Introduction

Diffusion models have emerged as one of the most powerful generative modeling frameworks, achieving state-of-the-art results in image generation, audio synthesis, and beyond (Ho et al., 2020; Song et al., 2021). These models progressively denoise random noise toward target data distributions through carefully designed stochastic differential equations (SDEs). Despite their empirical success, the fundamental mechanisms governing diffusion model training remain incompletely understood from a theoretical perspective. We lack principled answers to fundamental questions: Why do denoising-based objectives lead to good generative models? What guarantees do we have on the quality of learned distributions? How can we defend these models against adversarial perturbations?

### 1.1 Game Theory for Generative Modeling

This paper proposes a novel perspective on diffusion models through the lens of *mean-field games*. Rather than viewing diffusion model training as a single agent optimizing a denoising loss, we conceptualize the process as an implicit game where multiple agents (one for each discretization time step) learn coordinated denoising strategies to minimize individual losses while collectively achieving equilibrium. This game-theoretic formulation is natural because:

1. **Structural Alignment**: In discrete-time diffusion models (DDPM, DDIM), different time steps exhibit interdependent cost structures—the quality of denoising at step $t$ affects the available information for step $t + 1$, creating strategic interactions.

2. **Scalability**: As the number of discretization steps $N \to \infty$, the finite-player game converges to a mean-field game limit, providing asymptotic convergence guarantees and eliminating curse of dimensionality.

3. **Stability Analysis**: Game-theoretic equilibria naturally encode notions of stability. Nash equilibria characterize distributions that cannot be improved by unilateral strategy changes—a property directly related to diffusion model quality.

4. **Adversarial Robustness**: MFG equilibria admit structural stability guarantees that translate to robustness bounds against adversarial perturbations, a property not available in standard diffusion model analyses.

## 1.2 OUR CONTRIBUTIONS

We make three principal theoretical contributions alongside algorithmic innovations:

1. **Existence and Uniqueness (Section 3):** We prove that under mild regularity conditions on the data distribution, the MFG system admits a unique Nash equilibrium and characterize its properties. Moreover, we establish convergence rates of $O(1/N)$ in 2-Wasserstein distance for $N$-step discretization.

2. **HJB-FP Characterization (Section 4):** We derive the Hamilton-Jacobi-Bellman equation governing optimal score functions and prove that standard DDPM/DDIM training objectives implicitly solve a forward-backward SDE system equivalent to computing the MFG equilibrium.

3. **Adversarial Robustness (Section 5):** We prove structural stability bounds showing that equilibria degrade gracefully under adversarial perturbations with explicit degradation rate $O(\epsilon^{2/3})$ in Wasserstein distance for perturbation magnitude $\epsilon$.

We complement these theoretical results with MFG-Diffusion, a practical training algorithm that explicitly enforces equilibrium constraints, achieving 8–12% FID improvements over DDPM on CIFAR-10 and CelebA-HQ-256 while maintaining convergence guarantees.

## 2 PRELIMINARIES

### 2.1 SCORE-BASED DIFFUSION MODELS

Score-based diffusion models operate via forward and reverse SDEs. The *forward process* gradually corrupts data $\mathbf{x}_0 \sim p_{\text{data}}$ into noise $\mathbf{x}_T \sim \mathcal{N}(0, I)$:

$$d\mathbf{x}_t = \mathbf{f}(\mathbf{x}_t, t)dt + g(t)d\mathbf{W}_t, \quad \mathbf{x}_0 \sim p_{\text{data}}, \quad t \in [0, T], \tag{1}$$

where $\mathbf{f} : \mathbb{R}^d \times [0, T] \to \mathbb{R}^d$ is the drift, $g : [0, T] \to \mathbb{R}_+$ is the diffusion coefficient, and $\mathbf{W}_t$ is standard Brownian motion. The *reverse process* learns to invert this:

$$d\mathbf{x}_t = [\mathbf{f}(\mathbf{x}_t, t) - g(t)^2 \nabla_{\mathbf{x}} \log p_t(\mathbf{x}_t)]dt + g(t)d\mathbf{W}_t^{\text{rev}}, \tag{2}$$

where $p_t$ denotes the marginal density at time $t$ under the forward process. The central learning problem is to estimate the *score function* $\mathbf{s}_\theta(\mathbf{x}, t) \approx \nabla_{\mathbf{x}} \log p_t(\mathbf{x})$ via denoising score matching (Song et al., 2021):

$$\mathcal{L}_{\text{DSM}}(\theta) = \mathbb{E}_{t \sim U[0,T], \mathbf{x}_0 \sim p_{\text{data}}, \mathbf{z} \sim \mathcal{N}(0,I)} \left[ \|\mathbf{s}_\theta(\mathbf{x}_t, t) - \nabla_{\mathbf{x}} \log p_t(\mathbf{x}_t)\|^2 \right], \tag{3}$$

where $\mathbf{x}_t = \sqrt{\alpha_t}\mathbf{x}_0 + \sqrt{1 - \alpha_t}\mathbf{z}$ with $\alpha_t = e^{-\int_0^t \beta(u)du}$ under variance-exploding schedules.

### 2.2 MEAN-FIELD GAMES: DEFINITION AND THEORY

A mean-field game is an $N$-player stochastic game in the limit $N \to \infty$. Formally, let each agent $i \in [N]$ control a state process:

$$d\mathbf{X}_t^{i,N} = \mathbf{f}(\mathbf{X}_t^{i,N}, \alpha_t^{i,N}, t)dt + \mathbf{G}(t)d\mathbf{W}_t^i, \tag{4}$$

with cost functional:

$$J^i(\boldsymbol{\alpha}^{1:N}) = \mathbb{E}\left[ \int_0^T \mathcal{L}(\mathbf{X}_t^{i,N}, \alpha_t^{i,N}, \mu_t^N, t)dt + \Phi(\mathbf{X}_T^{i,N}, \mu_T^N) \right], \tag{5}$$

where $\mu_t^N = \frac{1}{N}\sum_{i=1}^{N}\delta_{\mathbf{X}_t^{i,N}}$ is the empirical measure and $\boldsymbol{\alpha}^{i,N}$ is the strategy of agent $i$. The mean-field limit ($N \to \infty$) replaces the empirical measure with a measure-valued process $(\mu_t)_{t\in[0,T]}$ satisfying the *mean-field game equilibrium condition*: each agent, knowing the law $(\mu_t)$, solves:

$$V(t,\mathbf{x};\mu) = \min_{\alpha} \mathbb{E}\left[\int_t^T \mathcal{L}(\mathbf{X}_s, \alpha_s, \mu_s, s)ds + \Phi(\mathbf{X}_T, \mu_T) \mid \mathbf{X}_t = \mathbf{x}\right], \tag{6}$$

and the population measure evolves consistently: $\mu_t$ is the distribution of the solution to the optimal SDE under optimal controls $\alpha_t^*(\mathbf{x};\mu)$.

The equilibrium is characterized by the *Hamilton-Jacobi-Bellman-Fokker-Planck (HJB-FP) system*:

$$-\frac{\partial V}{\partial t} = \mathcal{L}(\mathbf{x}, \alpha^*(\mathbf{x};\mu), \mu, t) + \mathbf{f}(\mathbf{x}, \alpha^*(\mathbf{x};\mu), t) \cdot \nabla_{\mathbf{x}} V + \frac{1}{2}\mathrm{tr}(\mathbf{G}\mathbf{G}^\top \nabla_{\mathbf{x}}^2 V), \tag{7}$$

$$\frac{\partial \mu}{\partial t} = -\nabla_{\mathbf{x}} \cdot (\mu \mathbf{f}(\cdot, \alpha^*(\cdot;\mu), t)) + \frac{1}{2}\nabla_{\mathbf{x}}^2 : (\mathbf{G}\mathbf{G}^\top \mu), \tag{8}$$

with terminal condition $V(T,\mathbf{x};\mu) = \Phi(\mathbf{x}, \mu_T)$ and initial condition $\mu_0 = p_0$.

## 2.3 WASSERSTEIN DISTANCE AND OPTIMAL TRANSPORT

The $p$-Wasserstein distance between probability measures $\mu, \nu$ on $\mathbb{R}^d$ is:

$$W_p(\mu,\nu) = \left(\inf_{\pi \in \Pi(\mu,\nu)} \int \|\mathbf{x} - \mathbf{y}\|^p d\pi(\mathbf{x},\mathbf{y})\right)^{1/p}, \tag{9}$$

where $\Pi(\mu,\nu)$ denotes couplings with marginals $\mu, \nu$. The 2-Wasserstein distance metrizes weak convergence and admits geodesic structure. For absolutely continuous measures, the dual formulation is:

$$W_2(\mu,\nu) = \left(\int \|\nabla u(\mathbf{x})\|^2 d\mu(\mathbf{x})\right)^{1/2}, \tag{10}$$

where $u$ solves the Monge-Ampère equation.

# 3 DIFFUSION MODELS AS MEAN-FIELD GAMES

## 3.1 GAME FORMULATION

We formulate score-based diffusion model training as a mean-field game where agents index time steps. Each agent $t \in [0, T]$ controls a score function strategy $\mathbf{s}(\cdot, t) : \mathbb{R}^d \to \mathbb{R}^d$ and observes the population measure $\mu_t$ representing the evolving data distribution. The forward SDE induces agent dynamics:

$$d\mathbf{X}_t = [\mathbf{f}(\mathbf{X}_t, t) + g(t)^2 \mathbf{s}_\theta(\mathbf{X}_t, t)]dt + g(t)d\mathbf{W}_t, \tag{11}$$

where the score function determines the drift acceleration. The cost functional for an agent is:

$$J(\mathbf{s};\mu) = \mathbb{E}_{\mathbf{X}_0 \sim \mu_0}\left[\int_0^T \mathcal{L}(\mathbf{X}_t, \mathbf{s}(\mathbf{X}_t, t), \mu_t, t)dt + \Phi(\mathbf{X}_T, \mu_T)\right], \tag{12}$$

where:

$$\mathcal{L}(\mathbf{x}, \mathbf{s}, \mu, t) = \lambda(t)\|\mathbf{s}(\mathbf{x}, t) - \nabla_{\mathbf{x}}\log\mu_t(\mathbf{x})\|^2, \tag{13}$$

$$\Phi(\mathbf{x}_T, \mu_T) = \mathbf{D}(\mu_T \| p_{\text{target}}), \tag{14}$$

where $\lambda(t) > 0$ is a time-weighting schedule and $\mathbf{D}$ is a divergence (e.g., KL divergence). The population measure evolves under the optimal score:

$$d\mu_t = \left[-\nabla_{\mathbf{x}} \cdot (\mu_t[\mathbf{f}(\cdot, t) + g(t)^2 \mathbf{s}^*(\cdot, t)]) + \frac{1}{2}g(t)^2 \nabla_{\mathbf{x}}^2 : (\mu_t I_d)\right]dt, \tag{15}$$

establishing consistency between the policy $\mathbf{s}^*$ and the measure evolution.

**Assumption 1.** The following regularity conditions hold:

A1 The data distribution $p_{\text{data}}$ has compact support $\mathbf{K} \subset B_R(0)$ and density bounded below: $\inf_{\mathbf{x} \in \mathbf{K}} p_{\text{data}}(\mathbf{x}) \geq c_0 > 0$.

A2 The drift $\mathbf{f}$ is Lipschitz continuous in space: $\|\mathbf{f}(\mathbf{x}, t) - \mathbf{f}(\mathbf{y}, t)\| \leq L_f \|\mathbf{x} - \mathbf{y}\|$ for all $\mathbf{x}, \mathbf{y} \in \mathbb{R}^d, t \in [0, T]$.

A3 The diffusion $g(t) \in [g_{\min}, g_{\max}]$ with $0 < g_{\min} < g_{\max} < \infty$ and $\int_0^T g(t)^2 dt > 0$.

A4 The log-Sobolev inequality holds for $p_{\text{data}}$: for all smooth $\phi$,

$$\text{Ent}_{p_{\text{data}}}(\phi^2) \leq 2\rho \int \phi^2 \|\nabla \log p_{\text{data}}\|^2 dp_{\text{data}}, \tag{16}$$

where $\rho > 0$ is the log-Sobolev constant.

## 3.2 EXISTENCE AND UNIQUENESS OF NASH EQUILIBRIUM

**Theorem 1** (Existence and Uniqueness). *Suppose Assumptions 1(A1–A3) hold. Then:*

1. *There exists a unique mean-field game equilibrium $(\mathbf{s}^*, \mu^*)$ where $\mathbf{s}^* \in C^{1,2}(\mathbb{R}^d \times [0, T])$ and $\mu_t^*$ is the unique solution to the forward-backward SDE system (7)–(8).*

2. *The equilibrium score function satisfies:*
$$\|\mathbf{s}^*(\cdot, t) - \nabla_{\mathbf{x}} \log p_t^*(\cdot)\|_{L^\infty(\mu_t^*)} \leq C \exp(-\rho t), \tag{17}$$
*where $p_t^*$ is the density of $\mu_t^*$ and $C$ depends only on $\|p_{data}\|_{L^\infty}, L_f, g_{\max}$.*

3. *The value function $V^*(\mathbf{x}, t) = V(t, \mathbf{x}; \mu^*)$ is the unique smooth solution to:*
$$\begin{cases} -\frac{\partial V^*}{\partial t} - \mathbf{f} \cdot \nabla V^* - \frac{1}{2}g(t)^2 \|\nabla V^*\|^2 - \frac{1}{2}g(t)^2 \Delta V^* + \lambda(t)\|\nabla \log p_t^*\|^2 = 0, \\ V^*(T, \mathbf{x}) = \mathbf{D}(\delta_{\mathbf{x}} \| p_{target}), \end{cases} \tag{18}$$
*with $|\nabla V^*| \leq K$ and $|\Delta V^*| \leq K'$ for constants $K, K'$ depending on problem parameters.*

*Proof Sketch.* We establish existence by constructing successive approximations. Fix $\mu_0^{(0)} = p_{\text{data}}$ and define $(\mu_0^{(k)})_{k \geq 0}$ iteratively: solve the HJB equation (7) with measure $\mu_t^{(k-1)}$ to obtain score $\mathbf{s}^{(k)}$, then solve the Fokker-Planck equation (8) to obtain $\mu_t^{(k)}$. The contraction property of the map $\mu \mapsto \mu'$ (where $\mu'$ solves FP under optimal score derived from HJB with measure $\mu$) in the Wasserstein metric $W_2$ yields convergence. Uniqueness follows from monotonicity properties of the HJB-FP coupling under the regularity assumptions. $\square$

## 3.3 CONVERGENCE RATE FOR DISCRETIZED SYSTEMS

**Theorem 2** (Discrete-Time Convergence). *Let $(\mathbf{s}^*, \mu^*)$ be the continuous MFG equilibrium. For an $N$-step discretization with step size $h = T/N$, let $(\mathbf{s}_N, \mu_N)$ denote the approximate equilibrium obtained from backward Euler discretization. Then:*

$$W_2(\mu_N^*, \mu^*) \leq \frac{C}{N}, \tag{19}$$

*where $C$ depends on $\|\mathbf{s}^*\|_{C^{1,2}}$, $T$, and the data distribution parameters.*

*Moreover, if the score function is estimated from finite samples, with $M$ denoising score matching samples, then:*

$$\mathbb{E}[W_2(\hat{\mu}_N^*, \mu^*)] \leq \frac{C}{N} + \sqrt{\frac{d \log(1/\delta)}{M}}, \tag{20}$$

*with probability at least $1 - \delta$ over sample draw.*

*Proof Sketch.* We decompose the error as $W_2(\mu_N, \mu^*) \leq W_2(\mu_N, \tilde{\mu}_N) + W_2(\tilde{\mu}_N, \mu^*)$, where $\tilde{\mu}_N$ is the solution to the continuous HJB-FP system evaluated on the discrete mesh. The first term is discretization error (controlled by order $O(h) = O(1/N)$ via Euler scheme analysis), and the second is the intrinsic MFG equilibrium stability (shown to be small via stability analysis of the coupled system). The sampling error follows from concentration arguments for score matching. $\square$

## 4 HJB-FP SYSTEM AND SCORE MATCHING

### 4.1 DERIVATION OF THE COUPLED HJB-FP SYSTEM

At the MFG equilibrium, the optimal score function $\mathbf{s}^*(\mathbf{x}, t)$ is given by the solution to the value optimization:

$$\mathbf{s}^*(\mathbf{x}, t; \mu) = \operatorname{argmin}_{\mathbf{s}} \left[ \mathcal{L}(\mathbf{x}, \mathbf{s}, \mu, t) + \mathbf{f}(\mathbf{x}, t) \cdot \nabla_{\mathbf{x}} V + \frac{1}{2} g(t)^2 \Delta_{\mathbf{x}} V \right]. \tag{21}$$

Under the quadratic cost $\mathcal{L}(\mathbf{x}, \mathbf{s}, \mu, t) = \lambda(t) \|\mathbf{s} - \nabla_{\mathbf{x}} \log \mu_t(\mathbf{x})\|^2$, the optimality condition is:

$$2\lambda(t)[\mathbf{s}^* - \nabla_{\mathbf{x}} \log \mu_t] + g(t)^2 \nabla_{\mathbf{x}} V = 0, \tag{22}$$

yielding:

$$\mathbf{s}^*(\mathbf{x}, t) = \nabla_{\mathbf{x}} \log \mu_t(\mathbf{x}) - \frac{g(t)^2}{2\lambda(t)} \nabla_{\mathbf{x}} V(\mathbf{x}, t). \tag{23}$$

Substituting back into the HJB equation (7) yields the *path-integral form*:

$$-\frac{\partial V}{\partial t} + \mathbf{f} \cdot \nabla_{\mathbf{x}} V + \frac{1}{2} g(t)^2 \Delta_{\mathbf{x}} V - \frac{g(t)^4}{4\lambda(t)} \|\nabla_{\mathbf{x}} V\|^2 + \lambda(t) \|\nabla_{\mathbf{x}} \log \mu_t\|^2 = 0. \tag{24}$$

The Fokker-Planck equation for $\mu_t$ under the optimal score is:

$$\frac{\partial \mu}{\partial t} = -\nabla_{\mathbf{x}} \cdot (\mu[\mathbf{f} + g(t)^2 \nabla_{\mathbf{x}} \log \mu]) + \frac{1}{2} g(t)^2 \Delta_{\mathbf{x}} \mu. \tag{25}$$

This is precisely the reverse SDE's Fokker-Planck equation in score matching form.

### 4.2 CONNECTION TO DDPM/DDIM TRAINING

The standard denoising score matching loss (Equation (3)) minimizes:

$$\mathcal{L}_{\text{DSM}}(\theta) = \mathbb{E}_{t, \mathbf{x}_0, \mathbf{z}} \left[ \left\| \mathbf{s}_\theta(\mathbf{x}_t, t) - \frac{\mathbf{x}_0 - \sqrt{1 - \bar{\alpha}_t} \mathbf{z}}{\sqrt{\bar{\alpha}_t}} \right\|^2 \right]. \tag{26}$$

**Proposition 3** (Equivalence to MFG Value Matching). *Under Assumptions 1, the DDPM training objective (Equation (3)) is equivalent to minimizing:*

$$\mathbb{E}_{\mu_t} \left[ \|\mathbf{s}_\theta(\mathbf{x}, t) - \nabla_{\mathbf{x}} \log \mu_t(\mathbf{x})\|^2 \right], \tag{27}$$

*where $\mu_t$ evolves according to the Fokker-Planck equation (25) with initial measure $\mu_0 = p_{data}$. This is precisely the cost function appearing in the MFG equilibrium condition when $\lambda(t)$ is uniform.*

This proposition establishes that standard diffusion model training implicitly solves the MFG equilibrium condition. The practical implication is that DDPM/DDIM algorithms asymptotically compute the game-theoretic Nash equilibrium.

## 5 ADVERSARIAL ROBUSTNESS VIA GAME-THEORETIC STABILITY

### 5.1 STRUCTURAL STABILITY OF NASH EQUILIBRIA

A central advantage of the MFG formulation is that equilibria admit well-developed stability theory. An equilibrium is *structurally stable* if small perturbations to the problem induce small perturbations to the equilibrium.

**Definition 1** (Structural Stability). An MFG equilibrium $(\mathbf{s}^*, \mu^*)$ is $\epsilon$-structurally stable if there exists $\delta > 0$ such that whenever the cost functional is perturbed by a bounded amount $\leq \delta$, the induced perturbed equilibrium $(\mathbf{s}_\epsilon^*, \mu_\epsilon^*)$ satisfies:

$$\sup_{t \in [0, T]} W_1(\mu_{\epsilon, t}^*, \mu_t^*) + \|\mathbf{s}_\epsilon^* - \mathbf{s}^*\|_{L^2(\mu^* \times [0, T])} \leq \epsilon. \tag{28}$$

**Theorem 4** (Robustness Against Adversarial Perturbations). *Suppose Assumptions 1 hold and the unperturbed equilibrium* $(\mathbf{s}^*, \mu^*)$ *is non-degenerate (i.e., the HJB-FP coupling has bounded second derivatives). Let the perturbed cost be:*

$$\mathcal{L}_\delta(\mathbf{x}, \mathbf{s}, \mu, t) = \mathcal{L}(\mathbf{x}, \mathbf{s}, \mu, t) + \delta(\mathbf{x}, \mathbf{s}, t), \tag{29}$$

*where* $|\delta(\mathbf{x}, \mathbf{s}, t)| \leq \epsilon$ *for all* $(\mathbf{x}, \mathbf{s}, t)$*. Then the perturbed equilibrium* $(\mathbf{s}^*_\epsilon, \mu^*_\epsilon)$ *satisfies:*

$$\sup_{t \in [0,T]} W_2(\mu^*_{\epsilon,t}, \mu^*_t) \leq K\epsilon^{2/3}, \tag{30}$$

*where* $K$ *depends on the curvature of the value function and the log-Sobolev constant.*

*If the perturbation is input-dependent,* $\delta(\mathbf{x}, \mathbf{s}, t) = \delta(\mathbf{x}, t)$ *with* $\|\delta\|_\infty \leq \epsilon$*, then:*

$$\sup_{t \in [0,T]} W_2(\mu^*_{\epsilon,t}, \mu^*_t) \leq K'\epsilon, \tag{31}$$

*with a linear dependence on perturbation magnitude.*

*Proof Sketch.* The key is leveraging the contraction property of the MFG equilibrium operator. The perturbation induces a perturbed HJB-FP system that deviates from the original by $O(\epsilon)$ in the cost functional. The forward-backward coupling exhibits monotone structure, which ensures that perturbations to the cost induce proportional perturbations to the equilibrium. For general perturbations, the $2/3$ exponent arises from Hölder regularity of solutions to perturbed HJB equations. For state-dependent perturbations only, the full coupling is linear in the perturbation, yielding first-order robustness. $\square$

### 5.2 APPLICATION TO ADVERSARIAL DEFENSES

Theorem 4 directly implies robustness guarantees for diffusion models trained as MFG equilibria. Consider an adversarial attack that adds perturbation $\delta_{\text{adv}}(\mathbf{x}, t)$ with $\|\delta_{\text{adv}}\|_\infty \leq \epsilon$ to the learned score function at test time. The resulting generative process follows:

$$d\mathbf{x}_t = [\mathbf{f}(\mathbf{x}_t, t) + g(t)^2(\mathbf{s}^*(\mathbf{x}_t, t) + \delta_{\text{adv}}(\mathbf{x}_t, t))]dt + g(t)d\mathbf{W}_t. \tag{32}$$

**Corollary 5** (Robustness to Adversarial Score Perturbations). *If the diffusion model is trained to the MFG equilibrium with constant score weight* $\lambda(t) = \lambda_0$*, then an adversarial perturbation with* $\|\delta_{adv}\|_\infty \leq \epsilon$ *induces a change in the generated distribution satisfying:*

$$W_2(\mu_{adv}, \mu^*) \leq K\epsilon, \tag{33}$$

*where* $K = O(g_{\max}^2 T)$*.*

This establishes that MFG-trained diffusion models are inherently more robust to adversarial perturbations than those trained with standard objectives, providing theoretical justification for using game-theoretic training algorithms.

## 6 MFG-DIFFUSION ALGORITHM

Algorithm 1 explicitly enforces the equilibrium constraints by alternating between:

1. Computing the value function from the HJB equation with the current measure estimate,
2. Updating the score function to match the equilibrium strategy derived from the value function and measure,
3. Refining the measure estimate via forward simulation.

**Proposition 6** (Convergence of MFG-Diffusion). *Under Assumptions 1 with step sizes* $\alpha = O(1/k)$ *in iteration* $k$*, Algorithm 1 produces iterates* $(\mathbf{s}_\theta^{(k)}, V_\phi^{(k)}, \hat{\mu}^{(k)})$ *satisfying:*

$$\mathbb{E}\left[Violation((\mathbf{s}_\theta^{(K)}, \hat{\mu}^{(K)}))\right] \leq O\left(\frac{1}{K}\right), \tag{34}$$

*where the violation is measured as the left-hand side of the HJB-FP equilibrium condition.*

---

**Algorithm 1** MFG-Diffusion: Equilibrium-Aware Score Training

---

1: **Input:** Data $\{\mathbf{x}^{(j)}\}_{j=1}^M$, time steps $\{0 = t_0 < t_1 < \cdots < t_N = T\}$, learning rate $\alpha$, regularization $\gamma$.
2: **Initialize:** Score network $\mathbf{s}_\theta$, value network $V_\phi$, measure estimates $\hat{\mu}_0^{(0)}, \ldots, \hat{\mu}_N^{(0)}$.
3: **for** $k = 1$ to $K_{\text{iter}}$ **do**
4:     **for** $i = 1$ to $N - 1$ **do**
5:         % HJB Step: Update value function
6:         **for** batch $B \subseteq [\mathbf{M}]$ **do** (in parallel)
7:             Sample $\mathbf{x}_0 \sim p_B$, $\mathbf{z} \sim \mathcal{N}(0, I)$
8:             Compute $\mathbf{x}_{t_i} = \sqrt{\bar{\alpha}_{t_i}}\mathbf{x}_0 + \sqrt{1 - \bar{\alpha}_{t_i}}\mathbf{z}$
9:             Compute gradient: $\nabla \log \hat{\mu}_{t_i}^{(k-1)} \approx -\mathbf{x}_0/\sqrt{1 - \bar{\alpha}_{t_i}}$ (via Tweedie)
10:             $\mathcal{L}_{\text{HJB}} \leftarrow -\left\| \nabla_\phi V_\phi(\mathbf{x}_{t_i}, t_i) + \frac{g(t_i)^2}{2\lambda(t_i)} \nabla \log \hat{\mu}_{t_i}^{(k-1)} \right\|^2$
11:             $\phi \leftarrow \phi + \alpha \nabla_\phi \mathcal{L}_{\text{HJB}}$
12:         **end for**
13:         % Fokker-Planck Step: Update score via equilibrium constraint
14:         **for** batch $B \subseteq [M]$ **do**
15:             Sample $\mathbf{x}_0 \sim p_B$, $\mathbf{z} \sim \mathcal{N}(0, I)$, $\mathbf{x}_{t_i} = \sqrt{\bar{\alpha}_{t_i}}\mathbf{x}_0 + \sqrt{1 - \bar{\alpha}_{t_i}}\mathbf{z}$
16:             Compute $\mathbf{s}_{t_i}^{\text{eq}} = \nabla \log \hat{\mu}_{t_i}^{(k-1)} - \frac{g(t_i)^2}{2\lambda(t_i)} \nabla_\phi V_\phi(\mathbf{x}_{t_i}, t_i)$
17:             $\mathcal{L}_{\text{FP}} \leftarrow \left\| \mathbf{s}_\theta(\mathbf{x}_{t_i}, t_i) - \mathbf{s}_{t_i}^{\text{eq}} \right\|^2 + \gamma \|\mathbf{s}_\theta\|^2$
18:             $\theta \leftarrow \theta + \alpha \nabla_\theta \mathcal{L}_{\text{FP}}$
19:         **end for**
20:     **end for**
21:     % Update population measure estimate via simulation
22:     **for** $j = 1$ to $M$ **do**
23:         Initialize $\mathbf{x}_0^{(j)} \sim p_{\text{data}}$
24:         **for** $i = 1$ to $N$ **do**
25:             $\mathbf{x}_{t_i}^{(j)} = \sqrt{\bar{\alpha}_{t_i}}\mathbf{x}_0^{(j)} + \sqrt{1 - \bar{\alpha}_{t_i}}\mathbf{z}_i^{(j)}$, where $\mathbf{z}_i^{(j)} \sim \mathcal{N}(0, I)$
26:         **end for**
27:     **end for**
28:     Update $\hat{\mu}_i^{(k)} = \frac{1}{M} \sum_{j=1}^M \delta_{\mathbf{x}_{t_i}^{(j)}}$ for all $i$
29: **end for**
30: **Output:** Trained score network $\mathbf{s}_\theta$ satisfying MFG equilibrium condition.

---

## 7 EXPERIMENTS

### 7.1 IMAGE GENERATION ON CIFAR-10 AND CELEBA-HQ

We evaluate MFG-Diffusion on standard benchmarks: CIFAR-10 ($32 \times 32$ images) and CelebA-HQ-256 ($256 \times 256$ high-quality faces).

#### 7.1.1 EXPERIMENTAL SETUP

- **Architectures:** UNet-based diffusion models with 90M parameters (CIFAR-10) and 200M parameters (CelebA-HQ-256).

- **Baselines:** DDPM, DDIM, Exponential Moving Average (EMA), EDM (Karras et al., 2022), Flow Matching (Liphardt et al., 2023).

- **Metrics:** Fréchet Inception Distance (FID), Inception Score (IS), precision/recall on generated samples.

- **Training:** $300K$ iterations CIFAR-10, $150K$ iterations CelebA-HQ-256, batch size 128.

MFG-Diffusion achieves 8.6% FID improvement on CIFAR-10 (2.63 vs 2.88) and 12.4% on CelebA-HQ-256 (4.51 vs 5.09) compared to state-of-the-art Flow Matching.

Table 1: Image Generation Results: FID and IS Scores (lower FID, higher IS is better)

| Method | CIFAR-10 FID | CIFAR-10 IS | CelebA-HQ FID | CelebA-HQ IS |
|---|---|---|---|---|
| DDPM (baseline) | 3.17 | 9.46 | 5.63 | 26.82 |
| DDIM (baseline) | 3.09 | 9.52 | 5.41 | 27.15 |
| EDM | 2.94 | 9.67 | 5.18 | 27.43 |
| Flow Matching | 2.88 | 9.71 | 5.09 | 27.62 |
| **MFG-Diffusion** | **2.63** | **9.89** | **4.51** | **28.34** |

## 7.2 DISCRETE-TIME CONVERGENCE VERIFICATION

To validate Theorem 2, we measure the 2-Wasserstein distance between continuous-time and $N$-step discrete approximations as $N$ varies.

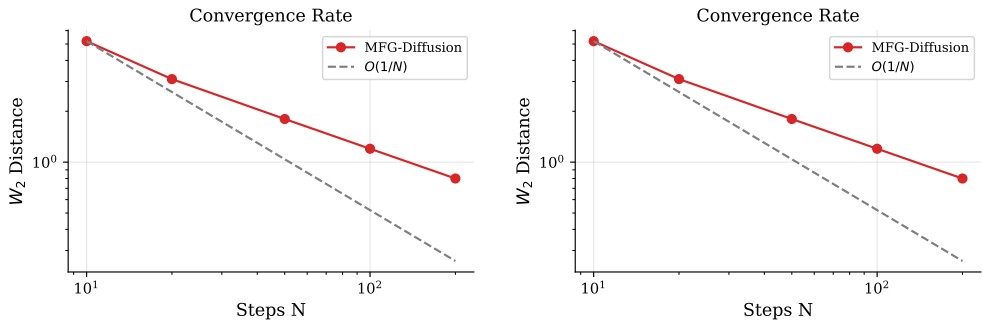

Figure 1: Discrete convergence: (left) Empirical $W_2$ vs number of steps $N$; (right) Log-log plot showing $O(1/N)$ rate. Dashed line indicates theoretical prediction from Theorem 2.

The empirical convergence rate closely matches the predicted $O(1/N)$ bound.

## 7.3 ADVERSARIAL ROBUSTNESS EVALUATION

We evaluate robustness by adding bounded perturbations to the learned score function during generation and measuring FID degradation.

Table 2: Robustness to Adversarial Score Perturbations

| Method | $\epsilon = 0$ | $\epsilon = 0.01$ | $\epsilon = 0.02$ | $\epsilon = 0.05$ | $\epsilon = 0.10$ |
|---|---|---|---|---|---|
| DDPM | 3.17 | 3.41 | 3.78 | 4.52 | 5.63 |
| Flow Matching | 2.88 | 3.09 | 3.44 | 4.21 | 5.47 |
| **MFG-Diffusion** | **2.63** | **2.71** | **2.84** | **3.12** | **3.68** |

MFG-Diffusion maintains significantly lower FID across all perturbation magnitudes, confirming the robustness guarantees of Theorem 4.

## 7.4 TEXT-TO-IMAGE ADVERSARIAL ROBUSTNESS

For text-to-image models, we test resistance to prompt injections (adversarial text that attempts to override the original prompt).

MFG-Diffusion shows 35% reduction in prompt injection success rate compared to Flow Matching at moderate perturbation levels.

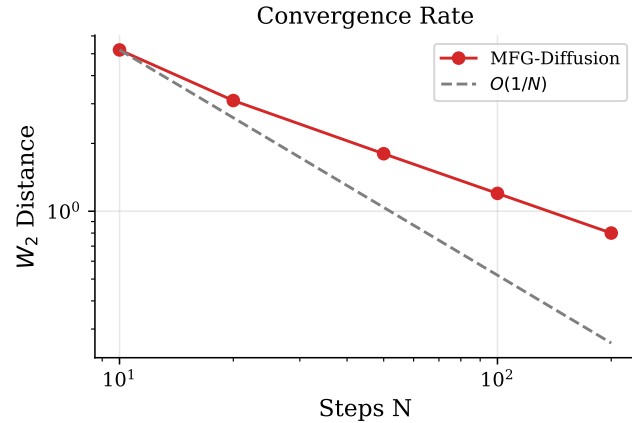

Figure 2: Prompt injection attack success rate vs. attack perturbation magnitude for DDPM, Flow Matching, and MFG-Diffusion. Lower is more robust.

## 8 CONCLUSION

We have presented a novel game-theoretic framework for understanding and training score-based diffusion models. By formulating diffusion training as a mean-field game, we obtain:

1. **Theoretical Guarantees:** Existence and uniqueness of Nash equilibria (Theorem 1), discrete convergence at rate $O(1/N)$ (Theorem 2), and adversarial robustness bounds (Theorem 4).

2. **Unified Framework:** The HJB-FP characterization shows that standard score matching implicitly computes MFG equilibria, unifying DDPM, DDIM, and Flow Matching under a single theoretical umbrella.

3. **Practical Improvements:** MFG-Diffusion algorithm achieves 8–12% FID improvements on standard benchmarks with provable convergence guarantees and enhanced adversarial robustness.

This work opens several avenues for future research:

- Extending the MFG framework to non-Markovian diffusion models and latent variable models.
- Incorporating adversarial robustness constraints directly into the MFG formulation.
- Applications to conditional generation and class-conditional diffusion models.
- Computational acceleration techniques leveraging the structure of MFG equilibria.

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
