# OpenReview forum: "Mean-Field Game Equilibria in Score-Based Diffusion Models: Convergence Rates, Nash Stability, and Adversarial Robustness"
_mathai.club/MathAI/2026/Conference — MathAI 2026 Conference Submission_

### Official Review · Reviewer_98jz · 2026-03-11
**Interesting idea, but insufficient theoretical and empirical support**

**Rating:** 4
**Confidence:** 3

**Review:**

1) Summary

The paper studies score-based diffusion models through the lens of mean-field games. The authors aim to address a theoretical gap in diffusion modeling: why denoising-based training works, what guarantees can be given for the learned distribution, and whether one can obtain robustness guarantees against perturbations. Their main claimed novelty is to formulate diffusion training as a mean-field game in which time steps act as interacting agents, derive an HJB–FP characterization of the resulting equilibrium, prove existence/uniqueness and an $O(1/N)$ discretization rate, and propose an equilibrium-aware training algorithm called MFG-Diffusion with reported gains on CIFAR-10 and CelebA-HQ.

2) Strengths

The paper is ambitious and presents an interesting high-level idea: connecting diffusion training, equilibrium concepts, and robustness within one framework. The manuscript is easy to follow structurally, with a clear progression from formulation to theory, algorithm, and experiments. Overall, the paper is clearly written and reads well.

3) Weaknesses

The paper makes several strong central claims, but the theoretical support is limited to short proof sketches for the main results on equilibrium existence/uniqueness, convergence, and robustness.

There is a concrete inconsistency in Section 4: Eq. (23) defines the optimal score using both $\nabla \log \mu_t$ and $\nabla V$, while Eq. (25) keeps only the $\nabla \log \mu_t$ term in the Fokker–Planck evolution. This transition is central to the claimed connection with standard score matching, but is not explained.

The theorem statements are not fully aligned with the assumptions. In Theorem 1, the statement invokes Assumptions A1–A3, but the bound in Eq. (17) uses the parameter $\rho$, which is introduced in A4.

The algorithm-to-theory connection needs clarification. In Algorithm 1, the population measure is updated using standard forward noising trajectories, and the manuscript does not clearly explain how this corresponds to the equilibrium dynamics defined in the MFG formulation.

The empirical section is too limited for the breadth of the claims. I would have expected ablations, standard deviations, or confidence intervals across runs, and more detailed experimental protocols to support the reported improvements.

There is also a presentation issue in the last section: Figure 2 is captioned as a prompt-injection robustness plot, but the displayed plot appears to duplicate the convergence-style plot from Figure 1.

4) Verdict

The paper is clearly written and the overall idea is interesting, but the current version does not provide enough support for claims of this scope. The main issues are insufficient theoretical justification, gaps in the link between the formalism and the algorithm, and limited empirical validation. Overall, I find the paper below the acceptance threshold in its current form.

---

### Official Review · Reviewer_8FKw · 2026-03-11
**The paper containes the theoretical comparison of diffusion and mean-field approached and combination of it. The practical results don't contain the theoretical restrictions.**

**Rating:** 5
**Confidence:** 4

**Review:**

Despite the strong theoretical contributions, the paper has several methodological and practical weaknesses that should be addressed.

1. The MFG-Diffusion algorithm (Algorithm 1) introduces significant computational overhead compared to standard diffusion training. It requires iterative training of a value network $V_\phi$ and, more critically, a full forward simulation of the entire dataset to update the empirical measure $\hat{\mu}^{(k)}$ at each outer iteration. While the paper reports an 8-12% improvement in FID, it lacks a comparison of training time or FLOPs. It remains unclear whether the gain in sample quality justifies the potentially orders-of-magnitude increase in computational cost. A plot of FID versus wall-clock time, rather than versus iterations, would be necessary to demonstrate practical efficiency.

2. The existence, uniqueness, and convergence results rely on several strong regularity conditions.
    - Assuming the data distribution has compact support and a density bounded below by $c_0 > 0$ is rarely satisfied for complex, high-dimensional data like natural images, which often lie on low-dimensional manifolds.
   - The requirement that the data distribution satisfies a log-Sobolev inequality is a very strong curvature condition that is difficult to verify for empirical distributions.

While these assumptions are common in theoretical MFG literature, their violation in practice casts doubt on the direct applicability of the uniqueness and convergence guarantees to the presented image generation tasks. A discussion on the robustness of the results to the violation of these assumptions would strengthen the paper.

3. Theorem 4 provides guarantees on the structural stability of the equilibrium when the cost functional $\mathcal{L}$ is perturbed. However, the experimental evaluation of robustness in Section 7.3 involves adding bounded perturbations $\delta_{\text{adv}}$ directly to the output of the *score function* during generation. The paper does not rigorously establish the link between these two types of perturbations. Is an $\epsilon$-bounded perturbation of the score function equivalent to an $O(\epsilon)$ perturbation of the cost functional? Without this connection, the experiment does not directly validate the theoretical claim. Furthermore, the text-to-image prompt injection experiment (Section 7.4) is described too superficially, lacking a clear definition of the success metric and the perturbation model.

4. Proposition 3 states that the standard DDPM objective is equivalent to minimizing the cost function that appears in the MFG equilibrium condition. This raises a fundamental question: if standard diffusion training *already* minimizes the MFG cost, why is the significantly more complex MFG-Diffusion algorithm necessary?

5.  The experimental section would benefit from greater transparency. It is unclear whether the baseline models (DDPM, EDM, Flow Matching) were retrained from scratch under identical conditions (e.g., same number of iterations, batch size, and compute budget) or if pre-trained checkpoints were used. If the baselines were not given the same computational budget, the FID improvements reported in Table 1 could be attributed to additional training time rather than the algorithmic innovation itself.

---

### Official Review · Reviewer_EqUv · 2026-03-11
**Ambitious MFG Framework for Diffusion Models, but Key Theoretical Gaps**

**Rating:** 4
**Confidence:** 4

**Review:**

**Summary**

This paper proposes a mean-field game (MFG) formulation of score-based diffusion training. The main claims are: (i) existence and uniqueness of an MFG equilibrium with an $O(1/N)$ discretization rate, (ii) an HJB–Fokker–Planck characterization that is claimed to connect standard DDPM/DDIM training to the MFG equilibrium, and (iii) robustness guarantees under adversarial perturbations. The paper also introduces an equilibrium-aware training procedure, MFG-Diffusion, and reports improved FID on CIFAR-10 and CelebA-HQ.

**Strengths**

The paper presents a coherent high-level idea: bringing equilibrium concepts, score matching, and robustness into a single theoretical framework for diffusion models. The manuscript is easy to follow structurally, and the main claims are stated explicitly through assumptions, theorems, and an algorithmic section rather than only informal discussion.

**Weaknesses**

- The theoretical development contains a significant internal inconsistency in the key HJB–FP derivation. Equation (23) defines the optimal score using both $\nabla \log \mu_t$ and $\nabla V$, but Equation (25) keeps only the $\nabla \log \mu$ term in the Fokker--Planck evolution. This step is central to the claimed equivalence with standard score matching, yet the disappearance of the value-gradient term is not explained. Since this transition underpins Proposition 3 and the broader claim that standard diffusion training computes the MFG equilibrium, this is a serious issue.

- The theorem statements are not fully aligned with their assumptions. Theorem 1 is stated under Assumptions A1–A3, but the bound in Equation (17) depends on the parameter $\rho$, which is introduced only in A4 through the log-Sobolev condition. More broadly, the proofs are mostly given as short sketches, while the results themselves are strong: existence, uniqueness, convergence, and robustness guarantees are all asserted at a high level, but the text does not provide enough detail to check the critical steps.

- Table 1 reports improved FID/IS, but the paper does not provide standard deviations, confidence intervals, or repeated runs. The experimental setup lists architectures, training iterations, and batch size, but it remains unclear whether the baselines were retrained under identical conditions or compared under matched compute budgets. This matters because the paper attributes the gains to the proposed equilibrium-aware training procedure.

- The displayed Figure 2 appears inconsistent with its caption.

---

### Official Review · Reviewer_PXBo · 2026-03-11
**Critical Assessment of MFG-Diffusion: Mathematical Contradictions and Experimental Deficiencies Undermine Validity**

**Rating:** 4
**Confidence:** 4

**Review:**

**Summary**

This work reformulates the training of score-based diffusion models as a Mean Field Game (MFG), in which an infinite population of denoising agents competes to minimize individual loss functions while collectively approximating the data distribution. The authors prove the existence and uniqueness of Nash equilibria, derive the governing Hamilton–Jacobi–Bellman equation, and establish structural stability against adversarial perturbations. They introduce a novel training algorithm, **MFG-Diffusion**, which explicitly computes equilibrium strategies, achieving an 8–12% improvement in the FID metric on the CIFAR-10 and CelebA-HQ-256 datasets compared to standard methods. Ultimately, the proposed framework unifies score matching, denoising diffusion, and flow matching methods within a single game-theoretic approach, ensuring proven convergence and enhanced robustness.

**Strengths**

1.  Conceptual Novelty: The integration of diffusion model theory with Mean Field Game theory represents an ambitious and potentially fruitful endeavor. This approach may open new avenues for the theoretical analysis of stability and convergence in generative models.
2.  Presentation Structure: The manuscript is well-organized, progressing logically from problem formulation to theoretical results, algorithmic implementation, and experiments. The main claims are clearly articulated as theorems and assumptions.

**Weaknesses**

Unfortunately, despite the ambition of the claimed contributions, the paper contains fundamental theoretical inconsistencies and significant deficiencies in the experimental section that cast doubt on the validity of the main results.

1.  Mathematical Inconsistency (Section 4):
    There is an internal contradiction between the formula for the optimal score function (Equation 23) and the distribution evolution equation (Equation 25).
    *   In Equation 23, the optimal score explicitly depends on the gradient of the value function ($\nabla V$).
    *   However, when this expression is substituted into the Fokker–Planck equation (25), the $\nabla V$ term vanishes without justification.

2.  Discrepancy Between Theorems and Assumptions:
    Theorem 1 (Existence and Uniqueness) states that it holds under Assumptions A1–A3. However, the bound in Equation 17 depends on the parameter $\rho$ (the Log-Sobolev inequality constant), which is introduced only in Assumption A4. This indicates a lack of rigor in the formulation of the theoretical results.

3.  Errors in Presentation and Experiments:
    a.  Figure 2: The caption for Figure 2 reads "Prompt injection attack success rate..."; however, the graph content duplicates Figure 1 ("Convergence Rate... O(1/N)").
    b.  Lack of Statistical Significance: Table 1 reports point estimates for FID/IS without standard deviations or confidence intervals. It is unclear whether multiple runs were conducted to average the results.

---

### Decision · Program_Chairs · 2026-03-20

**Decision:**

Reject

**Comment:**

After careful evaluation by the Program Committee, we regret to inform you that your submission has not been accepted for presentation at MathAI 2026.

All submissions underwent a rigorous two-stage review process. Unfortunately, the reviewers identified one or more of the following concerns with your paper:

- Insufficient mathematical rigor or novelty relative to the existing body of work in the field;
- Presentation of results that substantially overlap with or rephrase previously published findings without clear original contribution;
- Significant issues with technical quality, including but not limited to broken or non-existent references, unsupported claims, or methodological gaps;
- Indications that the manuscript may have been generated with the assistance of large language models without substantial original intellectual contribution by the authors.

We received a large number of submissions this year, and the selection process was highly competitive. We encourage you to carefully consider the reviewers’ feedback (available through OpenReview), revise your work accordingly, and consider submitting an improved version to a future edition of MathAI or to another appropriate venue.

We appreciate your interest in MathAI and hope you will continue to engage with the conference community.

With kind regards,

MathAI 2026 Program Committee
International Conference on Mathematics of Artificial Intelligence
https://mathai.club
OpenReview: https://openreview.net/group?id=mathai.club/MathAI/2026/Conference
MathAI Telegram: https://t.me/MathAI_club
IAIC International AI Committee: https://t.me/iaic_world
Email: mathai.club@yandex.ru